# Molecular and Cellular Mechanisms that Induce Arterial Calcification by Indoxyl Sulfate and P-Cresyl Sulfate

**DOI:** 10.3390/toxins12010058

**Published:** 2020-01-19

**Authors:** Britt Opdebeeck, Patrick C. D’Haese, Anja Verhulst

**Affiliations:** Laboratory of Pathophysiology, Department of Biomedical Sciences, University of Antwerp, 2000 Antwerpen, Belgium; britt.opdebeeck2@uantwerpen.be (B.O.); anja.verhulst@uantwerpen.be (A.V.)

**Keywords:** uremic toxins, arterial calcification, lipid metabolism, inflammation, coagulation, endothelial dysfunction, epigenetics

## Abstract

The protein-bound uremic toxins, indoxyl sulfate (IS) and p-cresyl sulfate (PCS), are considered to be harmful vascular toxins. Arterial media calcification, or the deposition of calcium phosphate crystals in the arteries, contributes significantly to cardiovascular complications, including left ventricular hypertrophy, hypertension, and impaired coronary perfusion in the elderly and patients with chronic kidney disease (CKD) and diabetes. Recently, we reported that both IS and PCS trigger moderate to severe calcification in the aorta and peripheral vessels of CKD rats. This review describes the molecular and cellular mechanisms by which these uremic toxins induce arterial media calcification. A complex interplay between inflammation, coagulation, and lipid metabolism pathways, influenced by epigenetic factors, is crucial in IS/PCS-induced arterial media calcification. High levels of glucose are linked to these events, suggesting that a good balance between glucose and lipid levels might be important. On the cellular level, effects on endothelial cells, which act as the primary sensors of circulating pathological triggers, might be as important as those on vascular smooth muscle cells. Endothelial dysfunction, provoked by IS and PCS triggered oxidative stress, may be considered a key event in the onset and development of arterial media calcification. In this review a number of important outstanding questions such as the role of miRNA’s, phenotypic switching of both endothelial and vascular smooth muscle cells and new types of programmed cell death in arterial media calcification related to protein-bound uremic toxins are put forward and discussed.

## 1. Introduction

During chronic kidney disease (CKD), uremic retention solutes accumulate in the bloodstream due to progressive kidney function loss. Three classes of uremic retention solutes exist: (i) low-molecular-weight water-soluble solutes (<500 Da), (ii) middle-molecular-weight solutes (>500 Da), and (iii) protein-bound solutes. This latter class is characterized by a limited dialytic removal due to the high molecular weight of the protein complexes that complicates their movement across the dialysis membrane [1]. Both indoxyl sulfate (IS) and p-cresyl sulfate (PCS) belong to the protein-bound uremic toxins and originate from protein fermentation in the intestine. The intestinal microbiota facilitates the breakdown of tyrosine/phenylalanine and tryptophan into, respectively, p-cresol and indole, which are absorbed and detoxified by oxidation and conjugation with sulfate [2]. In the bloodstream, IS and PCS bind to albumin, which implies that glomerular filtration does not take place and thus requires tubular transporter systems in the kidney to excrete these two protein-bound uremic toxins. Basolateral organic anion transporter 1 (OAT1) and 3 (OAT3), breast cancer resistance protein (BCRP) and multidrug resistance protein 4 (MRP4) belong to the IS and PCS tubular transport system [3]. Due to progressive kidney function loss, the concentration of IS and PCS increases with CKD stage in humans, as shown in Table 1 (adapted from Lin et al., *J. Food Drug Anal.*, 2019 [4]), ending up with levels of around 20-fold CKD stage 1. Moreover, free IS and PCS levels are 100-fold higher in pretreatment hemodialysis patients as compared to normal subjects [5]. There is, however, a high inter-individual variability, reflected by high standard deviations, which is also observed in other studies [6,7,8]. An explanation might be (i) alterations in colon microbiome attributed in part to dietary restrictions in CKD patients [9], (ii) modulation of IS and PCS transporters [10] and (iii) residual kidney function [11]. This inter-patient variability might influence the interpretation on the association between of IS and PCS concentrations and clinical outcomes. However, unbound PCS serum levels are suggested to hold a substantial predictive value for the survival in CKD patients [8]. Moreover, IS and PCS serum levels are also associated with cardiovascular disease and mortality [12,13]. Furthermore, Shafi et al. reported no association between total IS and PCS serum concentrations with cardiac death, sudden cardiac death, and first cardiovascular event [14]. These conflicting clinical associations demand for experimental study designs to unravel the role of IS and PCS in cardiovascular disease, which is crucial, as cardiovascular defects account for 50% of all deaths in CKD patients [15], having high serum IS and PCS levels.

## 2. Molecular Mechanisms by Which IS and PCS Induce Vascular Calcification

Arterial media calcification is a life-threatening disease that manifests in elderly and patients with chronic kidney disease (CKD) and diabetes mellitus. The disease phenotype is characterized by a passive and active deposition of calcium phosphate crystals in the media layer of the arterial wall that leads to arterial stiffness, which in turn induces hypertension, left ventricular hypertrophy, and impaired coronary perfusion. Arterial media calcification occurs already in the early stages of CKD, and more than half of the CKD patients on dialysis suffer from it [16,17]. Moreover, calcifications in the arterial wall are also present in children with CKD [18]. Moreover, CKD patients who also suffer from diabetes mellitus have a higher incidence of arterial calcification compared to nondiabetic hemodialysis patients [19]. Poor control of glucose levels is a predictor of arterial calcification in humans [19,20]. Our laboratory recently reported that both IS and PCS are important harmful vascular toxins, as they trigger moderate to severe arterial media calcification in CKD rats, which goes along with the activation of inflammation (i.e., acute phase response signaling pathway) and coagulation (i.e., intrinsic/extrinsic prothrombin activation pathway) pathways linked with increased circulating glucose levels and insulin resistance. These changes were even observed after four days of IS or PCS exposure, i.e., before arterial media calcifications had developed, indicating that the IS/PCS mediated upregulation of inflammation and coagulation precedes the vascular calcification process [21]. Additionally, in this study, escape from uremic-toxin-induced calcification was linked with liver X receptor and farnesoid X/liver X receptor signaling pathways, discussed more in detail below. This review focuses on these signaling pathways, as well as on the connection with endothelial dysfunction and the effects on microRNAs. 

### 2.1. Inflammation and Coagulation Signaling Pathways

The inflammatory acute phase response signaling pathway is a physiological host-defense mechanism to injury, including trauma, acute infection, and myocardial infarction [22]. In addition, a low-grade chronic acute phase response exists and is characterized by chronically elevated levels of acute phase proteins in response to metabolically triggered inflammation [23]. During CKD, the human body undergoes a permanent status of low-grade inflammation. The uremic retention solutes IS and PCS regulate inflammation in multiple cell types, including adipocytes, endothelial cells, macrophages, proximal tubular cells, and glial cells [24,25,26,27,28]. Inflammatory responses are associated with the development of arterial calcification [29]. Moreover, a recent study in 112 chronic hemodialysis patients correlated acute-phase proteins (i.e., C-reactive protein, ferritin, hepcidin, and albumin) with the development of abdominal aortic calcification [30]. In addition, the acute-phase proteins serum amyloid A and C-reactive protein have been reported not to be solely produced by hepatocytes, but also in the arterial wall, by vascular smooth muscle cells (VSMCs). Both proteins stimulate the phenotypic switch of VSMCs into bone-like cells through activation of the p38 MAPK pathway and oxidative stress pathways [31,32]. As mentioned above, our study also revealed that, after proteomic analysis of aortic tissue from either IS or PCS exposed CKD rats, coagulation pathways (intrinsic/extrinsic prothrombin activation pathways) play a central role in the arterial calcification process [21]. A close link between coagulation and inflammation exists as coagulation factors, such as fibrinogens and prothrombin, also belong to the acute phase proteins [22]. A recent study showed that IS induces platelet hyperactivity, with an elevated response to collagen and thrombin, through activation of the p38 MAPK and oxidative stress pathways [33]. Moreover, other studies reported an association between increased serum IS levels and thrombotic complications in CKD patients [33,34]. Then again, the thrombin–antithrombin complex level, used as a marker for thrombin formation in vivo, correlates with the presence and severity of coronary artery calcification [35,36]. An in vitro study of Kapustin et al. showed that Gla-containing coagulation factors (i.e., prothrombin and protein C and S) inhibit the vascular calcification process [37]. This might mean that the calcification-inducing effects of warfarin [38,39] can also be ascribed to its anti-coagulant actions, next to the fact that it prevents the production of the active form of matrix Gla protein, an important endogenous calcification inhibitor (warfarin inhibits carboxylation of Gla-proteins by inhibiting vitamin K recycling). All of these data suggest that coagulation and vascular calcification are interconnected and that the uremic toxins IS and PCS could play an important role herein. However, a recent multicenter randomized controlled trial showed that withdrawal of vitamin K antagonists in hemodialysis patients did not influence progression of arterial calcification progression after 18 months [40]. These conflicting data again indicate that the arterial calcification process is the result of a complex interplay between different pathological pathways. 

Interestingly, in our experimental study, a minority (3 out of 15) of CKD rats exposed to either IS or PCS did not develop arterial media calcification. This escape from uremic toxin-induced arterial calcification was related to liver X receptor and farnesoid X/liver X receptor signaling pathways, an important pathway in lipid metabolism. Activation of the liver X receptor (LXR) by the agonist GW3965 leads to anti-inflammatory actions in endothelial cells and macrophages by attenuation of the NF-kB pathway and IL-8 production [41,42]. Moreover, LXR activation inhibits cardiomyocyte apoptosis by restored mitochondrial membrane potential level and thus decreased reactive oxygen species (ROS) production [43]. These studies suggest that IS- and PCS-induced vascular calcification could be counteracted by preventing inflammation and oxidative stress events through the LXR pathway. This is further strengthened by results from our study in which, next to upregulation of acute phase response (inflammation), also oxidative stress pathways, i.e., Glutathione Mediated Detoxification and Glutathione Redox Reactions I, were observed in calcified aortic tissue of rats exposed to either IS or PCS for 7 weeks [21]. Furthermore, epigenetic involvement, often influenced by inflammation and oxidative stress, adds to the complexity by which uremic toxins trigger cardiovascular complications in CKD patients. IS has been associated with the regulating mammalian methyltransferase Set7/9, an epigenetic inducer of inflammatory genes, in VSMCs [44]. Moreover, data from an experimental rat study showed that IS exposure induces arterial thrombosis via decreased aortic levels of sirtuin 1, a class III histone deacetylase involved in oxidative stress [45]. We conducted a curated chemical and genomic/proteomic perturbagen matching analysis to predict upstream regulators that could be responsible for the observed changes in the arterial proteins linked to inflammation and coagulation signaling pathways. This analysis revealed a major role for altered energy and glucose metabolism, including elevated glucose levels and insulin receptor dysfunction [21]. This is interesting in view of the fact that diabetes increases the risk for vascular calcification in CKD patients [19]. Koppe et al. showed that chronic PCS exposure promoted insulin resistance and hyperglycemia in CKD mice [46]. Taken together, our study [21] suggests that IS and PCS stimulate the aortic media calcification via alterations in glucose metabolism, which in turn may stimulate inflammation, coagulation, and oxidative stress pathways in the aorta. It is worth noting that other studies have reported that the aryl hydrocarbon receptor, activated by IS, also regulates coagulation in VSMCs [47] and endothelial cells [48]. Interestingly, transient hyperglycemia also triggers Set7/9-mediated epigenetic changes in the promoter of NF-kB subunit p65, favoring overexpression of inflammatory genes [49]. Moreover, hyperglycemia favors the downregulation of sirtuin 1 in endothelial cells, and, by this, it upregulates vascular p66She gene transcription, which is implicated in mitochondrial ROS production and induction of apoptotic cell death. Moreover, metformin, an antidiabetic drug, has been reported to stimulate sirtuin 1 activation, which suppressed the increase of poly (ADP-ribose) polymerase (PARP) mediated mitochondrial ROS and thereby halted oxidative stress and inflammation in the retina of diabetic rats [50]. Remarkably, both metformin [51] and minocycline (PARP-inhibitor) [52] inhibit the development of arterial calcification in rats, pointing to the importance of oxidative stress and cell death events in the process of vascular calcification. Figure 1 gives a schematic overview of the interplay between inflammation, coagulation, and lipid metabolism and the role of epigenetics in IS and PCS-induced arterial calcification.

### 2.2. MicroRNAs, Upcoming Important Epigenetic Regulators in Uremic Toxins-Induced Vascular Calcification

MicroRNAs, small noncoding RNAs, are approximately 18–25 nucleotides long and regulate the protein expression of the target mRNA without affecting the gene sequence [53]. It has been well established that microRNAs play an important role in the vascular calcification process [54]. Protein-bound uremic toxins stimulate the expression of microRNA miR-92a in endothelial cells and, by this, suppress the gene expression of sirtuin 1, Krüppel-like factors 2 and 4, and endothelial nitric oxide synthase (eNOS) [55]. Interestingly, nitric oxide (NO) may be protective against arterial calcification, as in vitro NO inhibits murine VSMCs calcification and osteochondrogenic transdifferentiation via inhibition of TGFβ-induced phosphorylation of SMAD2/3 [56] and metformin halts arterial calcification through restoration of NO bioavailability (via the AMPK-eNOS-NO pathway) in rats [51,57]. As already mentioned above, metformin also increases the expression of sirtuin 1 [50] and thus might be able to counteract the IS/PCS-miR-92a triggered suppression of sirtuin 1 and, by this, decrease oxidative stress and inflammation events. 

Another microRNA, miR-29b, has been reported to influence the development of arterial calcification in VSMC cell cultures and 5/6th nephrectomized rats [58]. Moreover, a recent study showed that miR-29b was downregulated in human aortic smooth muscle cells in which IS induced calcification, and by this increased Wnt7b/β catenin signaling [59]. IS also promotes renal fibrosis by DNA hypermethylation of sFRP5, leading to activation of the Wnt/β catenin signaling pathway [60]. Interestingly, our laboratory found that sclerostin, a Wnt/β catenin inhibitor, might be linked to prevention of vascular calcification development [39]. Furthermore, uremic toxins also correlate with elevated serum levels of miR-126, miR-143, miR-145, and miR-223 [61]. This latter microRNA, when overexpressed, triggers the uptake of glucose via the glucose transporter GLUT4 in cardiomyocytes [62] and stimulates the arterial calcification process by acting on the VSMC phenotypic switch [63], which is further discussed in the next paragraph. For this reason, miR-223 might have played a role in the hyperglycemia seen in our previous study concomitantly with the development of IS/PCS induced aortic media calcification. This, in turn, could have stimulated inflammation, coagulation, and oxidative stress pathways in the aorta. Again, this points to an important role of hyperglycemia in the toxic effects of IS and PCS on the vasculature.

### 2.3. Novel Research Fields to be Explored

#### 2.3.1. Protein-Bound Uremic Toxins Influence Phenotypic Switch of Multiple Cell Types

The phenotypic transition of VSMCs into osteo/chondrogenic cells is a hallmark of the arterial calcification process [64]. Studies have shown that both IS and PCS are able to induce osteo/chondrogenic transdifferentiation of VSMCs by downregulation of smooth muscle genes (i.e., smooth muscle 22α, α-smooth muscle actin) and upregulation of bone-like genes (i.e., runx2, alkaline phosphatase, and osteopontin) [65,66]. This genotypic switching stimulates the VSMCs to produce calcifying exosomes, in which calcium phosphate crystals aggregate and, by this, mineralize the extracellular matrix [64]. Interestingly, IS and PCS also induce the transdifferentiation of other cell types, such as proximal renal tubular cells. When these cells are exposed to either IS or PCS, an epithelial to mesenchymal transition (EMT) occurs. During EMT, proximal renal tubular cells lose their adhesive characteristics with downregulation of E-cadherin in favor of mesenchymal fibroblast-like characteristics with upregulation of α-smooth muscle actin, fibronectin, N-cadherin, and vimentin. Several studies have shown that exposure to IS or PCS induces these phenotypic alterations, leading to glomerular sclerosis and interstitial fibrosis, with further stimulation of the progression of CKD [67,68,69]. Moreover, AST120, an oral spherical carbonaceous adsorbent, absorbs the precursors of IS and PCS and ameliorates the EMT process in renal tubular cells, and this is correlated with a decrease of serum IS levels [70]. Both TGFβ/Smad signaling and β-catenin signaling have been reported to act as main regulators of the uremic-toxin-induced EMT process in renal tubular cells [69,71]. In addition to EMT, endothelial to mesenchymal transition (EndMT), a subtype of EMT, involves the transdifferentiation of endothelial cells into mesenchymal stem-like cells which can differentiate further into multiple cell lineages: fibroblasts/myofibroblasts, osteoblasts/osteocytes, chondrocytes, and/or adipocytes [72]. Various studies have linked EndMT to the development of arterial calcification [73,74,75]. Its specific role in the calcification process, however, is not yet completely understood. It would be interesting to investigate whether IS and PCS are involved in the EndMT process, as (i) IS and PCS are known to promote arterial media calcification, and (ii) IS and PCS stimulate the EMT process in proximal tubular cells. Moreover, research so far predominantly focused on the transdifferentiation of VSMCs into bone-like cells, whilst the phenotypic transition of endothelial cells (EndMT) is often neglected and might be more important than generally thought [76]. Subsequently, in vitro experiments have demonstrated that both IS and PCS induce endothelial dysfunction, underlining even more that the endothelium plays a crucial role during uremic-toxin-induced arterial media calcification.

#### 2.3.2. The Endothelium, an Overlooked Structure in the Process of Uremic Toxin Induced Vascular Calcification

IS and PCS induce deleterious effects in the endothelium, including the inhibition of cell proliferation and wound healing, and the increase in oxidative stress responses, cell senescence, and the release of endothelial microparticles [77,78,79,80]. A prospective observational study in 41 CKD patients revealed that AST-120 treatment decreased serum IS levels, as well as the oxidized/reduced glutathione ratio [79]. Interestingly, a depletion of glutathione triggers ferroptotic events [81], a novel type of iron-mediated programmed cell death characterized by the accumulation of lipid peroxides. Moreover, IS-generated oxidative stress in human umbilical vein endothelial cells induces endothelial senescence [79,82]. Additionally, in other cell types, including proximal renal tubular cells and VSMCs, IS has been reported to accelerate the process of cell senescence, as evidenced by the upregulation of p53 activity [83,84]. Activation of this tumor-suppressor protein p53 in the cell initiates apoptosis, senescence, and ferroptosis [85]. Which of these cell fates eventually occurs depends on yet unknown mechanisms that direct allow p53 to selectively activate downstream targets. The involvement of cell senescence and apoptosis in arterial calcification has been reported [86,87]. To which extent ferroptosis is involved in the calcification process remains unknown and, therefore, could be a new research field that is worth being explored, in order to further unravel the mechanisms underlying the effects of IS and PCS, as well as other toxins on the development of vascular calcification. 

An additional challenging approach would consist in a further in-depth investigation at which extent the IS-induced release of microparticles derived from the endothelial cell membranes plays a role in the development of arterial calcification [88]. These endothelial microparticles (EMPs) have pro-inflammatory and pro-coagulant characteristics by interacting with monocytes. Moreover, EMPs accumulate different substances which influence the behavior of recipient cells. Buendia et al. reported that endothelial dysfunction favors the release of EMPs with a high content of calcium and bone morphogenetic protein-2, two important stimulators of the osteo/chondrogenic transdifferentiation of VSMCs [89]. In addition, microRNAs can be released via EMPs and influence endothelial function, i.e., through modulation of eNOS bioavailability, as mentioned above. Figure 2 gives an overview of the IS- and PCS-induced effects on VSMCs and endothelial cells.

## 3. Concluding Remarks

In conclusion, the protein-bound uremic toxins IS and PCS are considered to be harmful vascular toxins. Their vascular toxicity is associated with the upregulation of inflammation, coagulation, and oxidative stress pathways. Moreover, hyperglycemia seems to be an important player in these events. On the other hand, escape from IS/PCS-induced arterial media calcification was linked to activation of lipid metabolism. This suggests that the balance in glucose versus lipid metabolism could be crucial in uremic-toxin-induced vascular calcification.

## Figures and Tables

**Figure 1 toxins-12-00058-f001:**
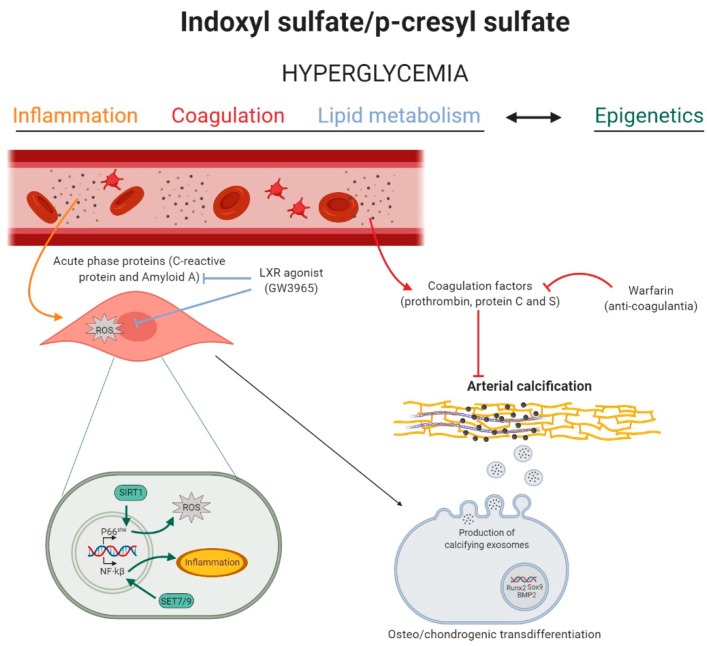
The role of inflammation, coagulation, lipid metabolism, and epigenetics in indoxyl sulfate and p-cresyl sulfate induced arterial calcification. High levels of indoxyl sulfate (IS) and p-cresyl sulfate (PCS) induce a state of hyperglycemia, which activates inflammation, coagulation, lipid, and epigenetic pathways in the vascular cell. Inflammation (yellow): acute-phase response proteins induce reactive oxygen species (ROS) production in the vascular smooth muscle cell, stimulating the phenotypic switch into osteo-/chondrogenic cells. Coagulation (red): circulating coagulation factors inhibit arterial media calcification, while the anti-coagulant warfarin stimulates the calcification process. Lipid metabolism (blue): the liver-X-receptor (LXR) agonist blocks the inflammation mediated ROS production and, by this, inhibits arterial media calcification. Epigenetics (green): IS and PCS induced hyperglycemia might trigger sirtuin 1 (SIRT1)- and Set7/9-mediated epigenetic changes in the promoter of, respectively, p66She gene and NF-kB subunit p65, favoring, respectively, mitochondrial ROS production and inflammation in the cell. Figure was created with BioRender.com.

**Figure 2 toxins-12-00058-f002:**
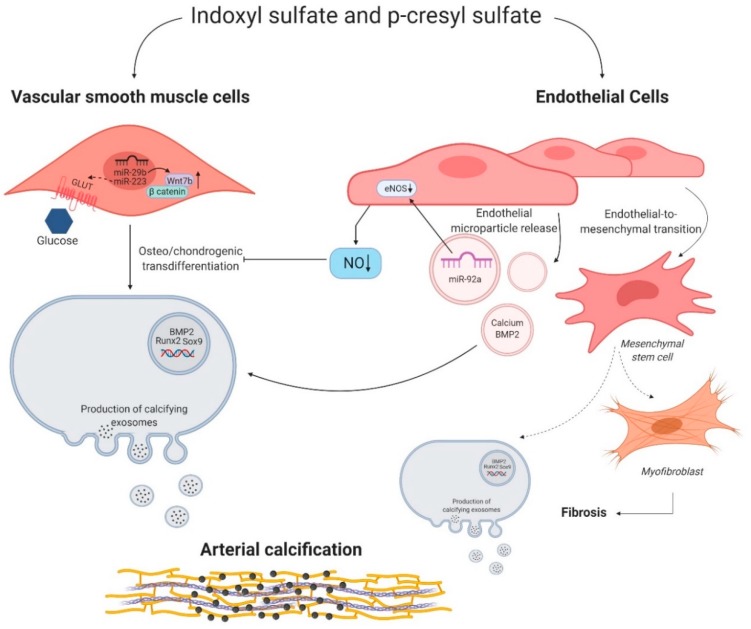
Indoxyl sulfate and p-cresyl sulfate induced molecular mechanisms in vascular smooth muscle cells and endothelial cells. Indoxyl sulfate (IS) and p-cresyl sulfate (PCS) influence the behavior of vascular smooth muscle cells (VSMCs) and endothelial cells. Right side: microRNA miR-29b and miR-223 favor the osteo/chondrogenic switch of VSMCs by promoting the expression of Wnt7b/β catenin signaling and potentially increasing the uptake of glucose via a glucose transporter (GLUT), respectively. Left side: IS and PCS stimulate endothelial microparticle release. These microparticles secrete microRNA miR-92a, calcium, and bone morphogenic protein 2 (BMP2), which in turn induce a phenotypic switch of VSMCs into osteo/chondrogenic cells directly or indirectly through influencing endothelial nitric oxide synthase (eNOS) and thus decreasing nitric oxide (NO) bioavailability. Moreover, IS and PCS could trigger the endothelial to mesenchymal transition of endothelial cells into osteo/chondrogenic cells and myofibroblast, stimulating arterial calcification and fibrosis. Figure was created with BioRender.com.

**Table 1 toxins-12-00058-t001:** Serum levels of albumin bound (total) and unbound (free) IS and PCS.

CKD	Stage 1	Stage 2	Stage 3	Stage 4	Stage 5
N	29	49	64	40	22
Total IS (mg/L)	1.03 ± 0.85	1.54 ± 1.11	2.22 ± 1.79	4.74 ± 4.34	18.21 ± 15.06
Total PCS (mg/L)	2.69 ± 4.34	4.42 ± 4.47	6.45 ± 7.12	16.10 ± 13.98	27.00 ± 17.66
Free IS (mg/L)	0.08 ± 0.06	0.11 ± 0.09	0.17 ± 0.13	0.49 ± 0.72	2.36 ± 2.64
Free PCS (mg/L)	0.15 ± 0.20	0.24 ± 0.29	0.36 ± 0.37	1.36 ± 2.58	2.38 ± 2.03

Data represent the mean ± standard deviation.

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
