# Peer review of "Molecular and Cellular Mechanisms that Induce Arterial Calcification by Indoxyl Sulfate and P-Cresyl Sulfate"

_toxins, 2020, doi:10.3390/toxins12010058_

Round 1
Reviewer 1 Report
General:
This review is initiated by a recent paper indicating that the activation of inflammation (i.e. acute phase response) and coagulation (i.e. intrinsic/extrinsic prothrombin) pathways are associated with hyperglycemia in CKD rats with arterial media calcification [15]. The authors provide insights about the roles of IS and PCS in the aortic media calcification via alterations of glucose metabolism, resulting in inflammation, coagulation and oxidative stress pathways in the aorta.
Specific comments:
There are two recent reviews (Am J Physiol Renal Physiol 2014, 307: F891; Am J Physiol Heart Circ Physiol 2017, 313: H1) regarding to molecular and cellular mechanisms of protein-bound uremic toxins induced by vascular calcification. Therefore, this review is better include what is known about protein-bound toxins and vascular calcification in these reviews. Title needs change according to the roles of IS and PCS in vascular calcification in the presence of hyperglycemia.Author Response
General:
This review is initiated by a recent paper indicating that the activation of inflammation (i.e. acute phase response) and coagulation (i.e. intrinsic/extrinsic prothrombin) pathways are associated with hyperglycemia in CKD rats with arterial media calcification [15]. The authors provide insights about the roles of IS and PCS in the aortic media calcification via alterations of glucose metabolism, resulting in inflammation, coagulation and oxidative stress pathways in the aorta.
Specific comments:
There are two recent reviews (Am J Physiol Renal Physiol 2014, 307: F891; Am J Physiol Heart Circ Physiol 2017, 313: H1) regarding to molecular and cellular mechanisms of protein-bound uremic toxins induced by vascular calcification. Therefore, this review is better include what is known about protein-bound toxins and vascular calcification in these reviews. Title needs change according to the roles of IS and PCS in vascular calcification in the presence of hyperglycemia.
We thank the reviewer for pointing out these two interesting reviews. We included both references in the revised manuscript (L.177 p6 and L.238 p7).
On the one hand, the manuscript relates indeed to the role of coagulation, inflammation and lipid pathways in IS/PCS-induced vascular calcification in the presence of hyperglycemia. However, on the other hand also focuses on the effects of IS and PCS on VSMCs versus endothelial cells and the role of miRNAs herein. We would prefer to keep the original title as we feel that it is concise, comprehensive and sufficiently covers the content of the manuscript.

Reviewer 2 Report
This is a good review that indoxyl sulfate (IS) and p-cresyl sulfate (PCS) are considered as harmful vascular toxins and their vascular toxicity is associated with up-regulation of inflammation, coagulation and oxidative stress pathways. Moreover, hyperglycemia seems to be an important player. I think it would be better to add more information about arterial calcification in DM in clinical settings.
Author Response
This is a good review that indoxyl sulfate (IS) and p-cresyl sulfate (PCS) are considered as harmful vascular toxins and their vascular toxicity is associated with up-regulation of inflammation, coagulation and oxidative stress pathways. Moreover, hyperglycemia seems to be an important player. I think it would be better to add more information about arterial calcification in DM in clinical settings.Indeed our study revealed that hyperglycemia may stimulate inflammation, coagulation and oxidative stress pathways in the aorta of IS and PCS exposed CKD rats. We therefore agree with the reviewer’s remark to include more information on the relationship between arterial calcification and diabetes mellitus in patients. We have revised this paragraph accordingly (L.69 p2, L143 p4)
Reviewer 3 Report
The manuscript “Molecular and cellular mechanisms that induce arterial calcification by indoxyl sulfate and p-cresyl sulfate” reviews the mechanisms leading to vascular calcification during CKD and the role of two major uremic toxins IS and PCS.
The authors stated that vascular calcifications contribute to mortality, this is not true. There is an association between vascular calcifications and CV mortality but no proof that decreasing vascular calcifications improved CV health so this must be stated and corrected in the abstract and the manuscript.
This sentence in the manuscript is a non-sense: “A complex interplay between inflammation-, coagulation- and lipid metabolism pathways as well as epigenetic factors is crucial in IS/PCS-induced arterial media calcification.” Inflammation, coagulation and lipid metabolism are the results of a relation between genes and environment (the so called epigenetic) so it is not true that there are the inflammation, coagulation in a part and epigenetic in another. This sentence must be change. In the last sentence of the abstract, the brackets must be removed.
L.37 p1, the authors must state that indole is also oxidized before to be sulfated.
44 p1, the authors report that the final concentrations is 20 fold in fact for IS it is 100 fold. This must be changed with adequate reference. https://www.ncbi.nlm.nih.gov/pubmed/24231664 46 p2 limit the variations of IS or PCS concentration to microbiota is false. Nutrition is important, restriction in proteins so in Aminoacid is linked to a decreased in IS and PCS concentration. Modulation of the transporters could also play an important role. Residual diuresis in dialysis is a major contributor to the IS and PCS concentration.L77 p2, the authors stated that Acute phase response, I imagine acute phase response of inflammation, is an host defense to injury, and they stated chronic inflammation, I Could not understand how chronic inflammation could induces APR. This must be explained or corrected.
L104, the authors must state that no data exist showing a reduction in vascular calcifications when using anticoagulant by citing this study https://www.ncbi.nlm.nih.gov/pubmed/31704740
L128 the authors suggested that activation of coagulation in CKD is only due to hyperglycemia. First less than the half of patients with CKD have diabetes and in these patients the CV risk is increased. Second other pathways was identified to explain activation of coagulation, mainly AhR by the Chitalia’s group in VSMC and Burtey’s group in Endothelial cells. So this must be corrected.
In the figure 1 the authors suggested that the main effects of warfarin is via coagulation but others important proteins playing an important role in calcification are modulated by VKA so this must be corrected.
The last part on the endothelium is too prospective with no experimental proof. EndMT was never study during CKD so it is very difficult to tell that it could play a role because we don’t know if it exists. Using data on proximal tubule is provocative when you know the poor quality of proximal cell in vitro. Same remark for ferroptosis and the role of endothelial cells in vascular calcification during CKD. We need data no extrapolation from various study. This part must be rewrite with proof or removed.
In conclusion, in a review the authors must limit their word to the proofs and reduced the amount of hypothesis.
Round 2
Reviewer 3 Report
The manuscript is greatly improvement and could be published.